# Molecular Mechanism of the *Grid* Gene Family Regulating Growth Size Heteromorphism in *Cynoglossus semilaevis*

**DOI:** 10.3390/ani15081130

**Published:** 2025-04-14

**Authors:** Yaning Wang, Yadong Chen, Yang Liu, Songlin Chen

**Affiliations:** 1College of Life Science, Qingdao University, Qingdao 266071, China; wwwwynyy66@163.com; 2State Key Laboratory of Mariculture Biobreeding and Sustainable Goods, Yellow Sea Fisheries Research Institute, Chinese Academy of Fishery Sciences, Qingdao 266071, China; chenyd@ysfri.ac.cn (Y.C.); liuyang@ysfri.ac.cn (Y.L.); 3Laboratory for Marine Fisheries Science and Food Production Processes, Qingdao Marine Science and Technology Center, Qingdao 266237, China

**Keywords:** GWAS, transcriptome, gh-igf1 growth axis, *cynoglossus semilaevis*

## Abstract

*Cynoglossus semilaevis* is a crucial fish for commercial aquaculture in China. However, females grow 2–4 times faster and larger than males, leading to significant individual growth differences. The species exhibits ZZ/ZW sex determination, with ZW females capable of undergoing sex reversal to become ZW physiological males. This sex reversal is heritable, causing more males in subsequent generations and increasing aquaculture costs. Addressing this issue involves identifying the genes responsible for growth variability. A genome-wide association study on the different body weight of fishes was executed to identify single nucleotide polymorphisms linked to the regulation of growth phenotypes. We discovered the *grid* gene, which suggests that it is related to growth. The *grid* gene family includes *grid1* and *grid2*, with a negative correlation between their expression and fish size. Males showed slightly higher expression levels of *grid* genes, and the *grid2* expression had a stronger correlation with fish size than *grid1*. Silencing the *grid* genes in the testicular cell line resulted in down-regulated igf1 and up-regulated gh, indicating that the *grid* gene family may influence the gh-igf1 axis and growth. We hypothesize that the *grid* gene family significantly affects the growth of *Cynoglossus semilaevis*, providing a basis for further research into size differences in this species.

## 1. Introduction

*Cynoglossus semilaevis*, a member of the Tongue Soleidae family, is extensively distributed in China’s Yellow, Bohai, and Yellow Sea regions [1]. This species is prized for its delectable, high-protein flesh, making it a vital fish for commercial aquaculture in China and a nutritious seafood option [2]. However, significant individual differences in the growth and development of *C. semilaevis* are evident, primarily manifested as sex-size dimorphism (SSD), whereby females grow 2–4 times faster and larger than males in both growth rate and body weight [3,4]. The species exhibits ZZ/ZW sex determination, with ZW females capable of undergoing sex reversal to become ZW physiological males (pseudo-males) under specific conditions like high temperatures [5]. This sex reversal is heritable, leading to an increasing proportion of males over generations, which raises aquaculture costs [5,6]. Researchers are currently focusing on identifying key genes that govern growth size heterosis in *C. semilaevis* from a sex differentiation perspective, aiming to enhance the size of male fish or reduce the proportion of males, thereby improving aquaculture efficiency [7,8,9]. Solving the growth variation issue in *C. semilaevis* hinges on identifying the genes responsible for this variability, offering new research directions for overcoming industry bottlenecks.

We conducted a GWAS analysis using the weight of 350 female individuals as the phenotypic trait. By identifying SNP loci related to weight, we discovered the *grid* (glutamate ionotropic receptor delta type subunit 1 gene) gene, which suggests that it is related to growth, and therefore, to the growth of tongue sole. The *grid* gene family comprises the *grid1* and *grid2* genes, encoding the GluD1 and GluD2 proteins, respectively, which form the δ (GluD) receptor [10]. The δ receptor belongs to a subfamily of ionotropic glutamate receptors (iGluRs), which are ligand-gated ion channels primarily involved in neural signaling. These receptors are abundantly located at the postsynaptic terminals of neurons and play crucial roles in fundamental neuronal processes such as learning, memory, long-term potentiation/depression, and synaptic plasticity [11]. In addition to AMPA, NMDA, and kainate receptors, δ receptors are integral to excitatory synaptic organization and function [12]. The GluD protein also modulates NMDA receptors and inhibits excitatory signaling between neurons, playing a significant role in neuroendocrine regulation and promoting growth hormone secretion [13]. This protein correlates with the growth and development of individual organisms, controlling hormone secretion and reproductive functions of the anterior pituitary gland [14,15]. The *grid1* gene, predominantly expressed in the adult mouse brain, participates in the glutamate signaling pathway and directly regulates the gonadotropin-releasing hormone (GnRH) [16]. The *grid2* gene is crucial for early developmental processes in mammals and other vertebrates, including migration regulation essential for embryogenesis [17,18]. *Grid2* is mainly found in the cerebellum of adults or mice, and its gene deletion results in developmental delays in humans [19]. However, the impact of the *grid* gene family on marine fish growth remains unreported.

This study aims to further investigate the relationship between growth-discrepant individuals and the *grid* gene family using genome-wide association analysis (GWAS) and transcriptome sequencing technologies. Phylogenetic tree construction and evolutionary analyses were conducted to elucidate evolutionary relationships and predict potential functions. Real-time quantitative PCR (qPCR) was employed to detect the expression patterns of the *grid* gene family in various tissues, establishing correlations with individual size. Additionally, the *grid* gene family’s expression was manipulated using small interfering RNA (siRNA), and the molecular mechanisms underlying its influence on growth were explored by analyzing the transcriptome post-interference. This research aims to provide reference data for studying the function and regulatory mechanisms of individual growth variation in *C. semilaevis*.

## 2. Materials and Methods

### 2.1. Experimental Fish and Tissue Collection

The experiment took one-year-old *Cynoglossus semilaevis*. Samples of *Cynoglossus semilaevis* were sourced from Haiyang City, Shandong Province, under the auspices of the Yellow Sea Aquatic Co. (Qingdao, China). Tissues including liver, kidney, spleen, intestine, brain, heart, muscle, skin, gonads, and gill filaments were collected and rapidly cryopreserved using liquid nitrogen, followed by storage at −80 °C for RNA extraction. Concurrently, caudal fin samples were clipped and preserved in anhydrous ethanol (Code No: 10009218, Sinopharm Chemical Reagent Co., Shanghai, China) for DNA extraction and subsequent genetic sex identification.

### 2.2. Experimental Program

The experiment was categorized into three groups based on the weight and length of the fish samples. The first group comprised fish of varying sexes and sizes [females: (30.8 ± 2.5) cm, (190.3 ± 10.2) g; males: (21.8 ± 1.8) cm, (63.5 ± 11.5) g]. The second group included one-year-old fish of the same sex but different sizes [large females: (43.0 ± 4.2) cm, (501.2 ± 86) g; small females: (35.6 ± 2.1) cm, (253.5 ± 38.5) g]. The third group consisted of fish of the same size but different sexes [females: (12.5 ± 1.1) cm, (11.8 ± 3) g; males: (12.1 ± 0.9) cm, (11.1 ± 2.1) g]. Five fish from each group were sampled as duplicates.

### 2.3. Total RNA Extraction and cDNA Synthesis

RNA extraction from tissue samples was conducted using the RNA Extraction Kit [Code No: Y1217, Tiangen Biochemical Technology (Beijing) Co., Beijing, China]. The concentration and quality of the RNA were assessed with the Ultra-Micro Nucleic Acid Detector (Puitton), and RNA integrity was verified by 1% agarose gel electrophoresis. High-quality RNA was then reverse transcribed to synthesize cDNA using the HiScript III 1st Strand cDNA Synthesis Kit (+gDNA wiper) (Code No. R312, Nanjing Novozymes Biotechnology Co., Nanjing, China).

### 2.4. Sequence Analysis and Comparison of Grid Gene Families

A genome-wide association study (GWAS) was executed to identify single nucleotide polymorphisms (SNPs) linked to the regulation of growth phenotypes. We conducted a preliminary GWAS analysis on 350 half-smooth tongue sole from 25 families. The study population had a total length ranging from 23.0 to 47.2 cm, with an average of 34.7 cm, and a weight range from 77 to 799 g, with an average of 308.6 g. There was a significant individual variation. Genomic DNA was extracted from the tail fin of 350 individual samples with different body weight using the TIANamp Marine Animal DNA Kit (Tiangen Biotech, Beijing, China). Post quality assessment, the qualified DNA underwent whole-genome resequencing via T7 for genotyping. Genotype data underwent whole-genome quality control post-resequencing. SNPs with a call rate below 90%, minor allele frequency below 5%, or significant deviation from the Hardy–Weinberg Equilibrium (*p* < 10^−5^) were excluded. Missing genotypes were imputed using Beagle 3.31. Post QC and imputation, the remaining markers were consolidated into a genotype dataset.

Based on the genomic data of *Cynoglossus semilaevis* (GCA_000525025.1) and the mRNA sequences for *grid1* (XM_008321593) and *grid2* (XM_008336887), we analyzed these sequences using the NCBI website (https://www.ncbi.nlm.nih.gov/tools/primer-blast/index.cgi?ORGANISM=244447&INPUT_SEQUENCE=) (accessed on 20 October 2024) to design PCR primers, with the primer sequences detailed in Table 1. The primers were synthesized by Beijing RuiBo Xingke Biotechnology Co., Ltd., Beijing, China. The coding sequences (CDS) of the *grid1* and *grid2* genes of *Cynoglossus semilaevis* were amplified using TOYOBO’s KOD DNA polymerase (Code No.: KMM-201, Toyobo, Tokyo, Japan). cDNAs were synthesized from RNA extracted from the gonadal tissues of *Cynoglossus semilaevis* and used as templates. The PCR reaction utilized primers *grid1*-F/R and *grid2*-F/R, respectively, in a total reaction volume of 20 µL consisting of 10 µL KOD enzyme, 1 µL cDNA template, 0.5 µL each of *grid1*-F/R (for *grid1*) or *grid2*-F/R (for *grid2*), and 8 µL ddH2O. The PCR amplification protocol was set as follows: initial denaturation at 98 °C for 5 min, 35 cycles of 98 °C for 10 s, 59 °C for 5 s, 68 °C for 35 s, and a final extension at 72 °C for 5 min, followed by holding at 4 °C. The PCR products were analyzed via 1% agarose gel electrophoresis. The target bands were excised, purified using a gel extraction kit (Code No: DC301, Nanjing Novozymes Biotechnology Co., Nanjing, China), and sequenced by Qingdao Kinko Biotechnology Co., Qingdao, China.

Successfully amplified cDNA sequences were compared and analyzed using BLAST (http://blast.ncbi.nlm.nih.gov) (accessed on 20 October 2024) for sequence comparison and coding sequence prediction in the NCBI database. The molecular weight and theoretical isoelectric point were predicted using the ExPASy website (https://web.expasy.org/protparam/) (accessed on 20 October 2024), and protein domain predictions were conducted with the SMART tool (http://smart.embl-heidelberg.de/) (accessed on 20 October 2024). Signal peptide prediction was carried out using SignalP 4.1 (https://services.healthtech.dtu.dk/service.php?SignalP-4.1) (accessed on 20 October 2024), and transmembrane region analysis was performed using TMHMM 2.0 (https://services.healthtech.dtu.dk/services/TMHMM-2.0/) (accessed on 20 October 2024). Phylogenetic tree construction was executed using MEGA 6.0 software.

### 2.5. Genomic DNA Extraction and Genetic Characterization

For DNA extraction, the caudal fin tissue was clipped and processed using the Animal Genomic DNA Extraction Kit (Magnetic Beads) (Code No: 0201151, Luoyang Aisen Biotechnology Co., Luoyang, China). DNA concentration was measured using the Ultra-Micro Nucleic Acid Detector, and DNA quality was evaluated by 2% agarose gel electrophoresis. The DNA of the half-smooth tongue sole was amplified using Premix Taq (Ex Taq Version 2.0 plus dye) (Code No. RR902A, TAKARA, Osaka, Japan). The PCR reaction utilized the extracted DNA as a template along with sex-specific primers, sex-F/R, as detailed in Table 1. The PCR reaction mixture totaled 20 µL: KODase 10 µL, DNA template 1 µL, sex-F 0.5 µL, sex-R 0.5 µL, and ddH2O 8 µL. The PCR amplification protocol was as follows: initial denaturation at 95 °C for 3 min, followed by 35 cycles of 95 °C for 15 s, 60 °C for 15 s, and 72 °C for 30 s, with a final extension at 72 °C for 5 min. The sex identification was confirmed using 2% agarose gel electrophoresis.

### 2.6. QPCR Detection of Gene Expression

To quantify gene expression levels in various tissues, qPCR experiments were conducted. Specific primers for the quantitative detection of *grid1* and *grid2* genes were designed using the NCBI website and synthesized by Qingdao Kengke Biotechnology Co., Qingdao, China. For the qPCR, cDNA templates synthesized from RNA extracted from different tissues were used. The total qPCR reaction volume was 20 µL, comprising 10 µL THUNDERBIRDTM Next SYBR^®^ qPCR Mix (Code No: QPX-201, TOYOBO), 1 µL cDNA, 0.3 µL each of forward and reverse primers, and 8.4 µL ddH2O. Each sample was run in four replicates. The qPCR reactions were performed on a real-time fluorescence quantitative PCR analyzer (Code No. FQD-96C, BIOER, Hangzhou, China), with the following cycling conditions: initial denaturation at 95 °C for 30 s, followed by 40 cycles of 95 °C for 5 s and 60 °C for 34 s, and a melting curve analysis performed using the default program. β-actin served as the internal reference gene, and the relative gene expression levels were calculated using the 2^−ΔΔCt^ method. Data were analyzed by one-way ANOVA and *t*-test using IBM SPSS Statistics 26 software, and differences were considered significant when *p* < 0.05.

### 2.7. Grid1 and Grid2 siRNA Interference

The *Cynoglossus semilaevis* testicular cell line was cultured in 25 cm^2^ flasks using 20% L15 medium (Code No: LA9510, Beijing Solebo Technology Co., Beijing, China) at a constant temperature of 25 °C. Once optimal cell growth was observed and the cell density reached between 80 and 90%, the cells were subjected to trypsin digestion and subsequently resuspended in 20% L15 medium. Following resuspension, 1 mL of the medium was aspirated from each well and uniformly transferred into a 12-well plate for overnight culture at 25 °C, ensuring complete cell adhesion to the well walls.

For transfection, the riboFECT^TM^ CP Transfection Kit (Code No. C10511-05, Guangzhou RuiBo Biotechnology Co., Guangzhou, China) was utilized, and specific siRNAs targeting the *grid1* and *grid2* genes were designed and synthesized by Sangon Bioengineering (Qingdao, China). According to the kit’s protocol, the following procedure was executed: 60 μL of 1 × Buffer, 3 μL of cp reagent, and 4 μL of siRNA mixture were gently added to each well of a 12-well plate. The plate was then slightly shaken to ensure thorough mixing of the premix with the medium and incubated at room temperature for 10 min before being transferred to a 25 °C incubator for further incubation. This setup included three replicates per condition. After 24 h of incubation, the transfection efficiency of siRNA was evaluated using an electron fluorescence microscope (Code No.: IX73P1F, OLYMPUS, Tokyo, Japan) with transfected Cy3-siRNA serving as the control group. After 48 h, cells from each transfection site were collected using TRIzol, and cellular RNA was extracted and reverse transcribed into cDNA. The level of gene silencing by each siRNA was quantified via qPCR, with NC-siRNA used as the control.

### 2.8. Transcriptome Analysis

The siRNA sites that successfully interfered with the *grid1* and *grid2* genes were selected for further experiments with *Cynoglossus semilaevis* gonadal cells, with three replicates per group. The RNA from cells of each transfected site was collected using TRIzol and labeled as follows: *grid1*-siRNA 1, 2, 3 experimental group and *grid1*-NC 1, 2, 3 control group; *grid2*-siRNA1, 2, 3 experimental group and *grid2*-NC1, 2, 3 control group. The corresponding cellular RNA was then reverse transcribed into cDNA, and the interference effects of the siRNA sites were confirmed by qPCR. Subsequently, the extracted cellular RNA was sent to the Shanghai Meiji Biomedical Technology Co. for transcriptome sequencing analysis in order to further investigate the molecular implications of the siRNA-mediated gene silencing.

## 3. Results

### 3.1. GWAS Analysis Confirmed the Growth-Related Genes

Through whole-genome resequencing of the *Cynoglossus semilaevis* genome and a subsequent genome-wide association study (GWAS) with phenotypic traits of body length and weight, a total of 1,536,597 SNP markers were used for GWAS analysis after whole-genome resequencing and quality control of the sequencing results. The GWAS results show that the highest number of trait-associated SNP loci were detected on the sex chromosomes (Z and W), and the highest phenotypic variation rate for female growth traits was observed. The upstream and downstream genes of these SNPs include genes such as akap9, anapc2, bcas3, cdca8, chst8, diaph2, fam168a, *grid2*, hip1, idh2, itgb6, lamb1, manba, mapk15, msto1, mtbp, and mthfd1 (Figure 1).

### 3.2. Cloning and Sequence Analysis of the Grid Gene Family

#### 3.2.1. Grid Gene Family Sequence Analysis and Prediction

The successfully cloned *Cynoglossus semilaevis grid1* gene sequence spans a length of 3525 bp, featuring an open reading frame (ORF) of 3042 bp that encodes 1013 amino acids (aa). The predicted molecular weight is 112,854 Da, and the theoretical isoelectric point (pI) is 6.11. This gene contains three transmembrane regions (Figure 2A). Additionally, a signal peptide is present with a cleavage site between amino acids 20 and 21 (Figure 3A). A detailed protein structural domain analysis is provided (Figure 4A).

Similarly, the cloned *Cynoglossus semilaevis grid2* gene sequence measures 5679 bp in length, encompassing an ORF of 2802 bp that encodes 933 amino acids (aa). The predicted molecular weight is 105,109 Da, and the theoretical pI is 6.24. This gene also includes three transmembrane regions (Figure 2B) and a signal peptide with a cleavage site between amino acids 18 and 19 (Figure 3B). The protein structural domain analysis is illustrated (Figure 4B).

#### 3.2.2. Evolutionary Analysis of Grid Gene Families

Phylogenetic analysis utilized the *grid1* and *grid2* gene sequences obtained from sequencing results to construct a phylogenetic tree with gene sequences from other species. This analysis revealed that *Cynoglossus semilaevis* clusters closely with zebrafish, indicating the closest affinity, while forming a more distant cluster with other vertebrates, consistent with evolutionary characteristics (Figure 5).

### 3.3. Differential Expression of the Grid Gene Family Across Tissues

Quantitative PCR (qPCR) was employed to explore the correlation of the *grid* gene family with individual size and sex in *Cynoglossus semilaevis*. The expression of the *grid1* gene in testicular tissues of smaller fish was approximately 2.5 times than in larger fish of the same sex (Figure 6A). In different sexes of the same size, the testicular expression was about 1.4 times higher than ovarian expression (Figure 6B). Across different sizes and sexes, the expression level in testicular expression was about 10.4 times higher than that in ovarian expression (Figure 6C).

For the *grid2* gene, expression in the testicular tissue of smaller fish was approximately 3.6 times higher than in larger fish of the same sex (Figure 6D). In the same sex and different sizes, the testicular expression was about 1.2 times higher than ovarian expression (Figure 6E). Across different sizes and sexes, the expression level in testicular was approximately 196.7 times higher than that in ovarian (Figure 6F).

### 3.4. Detection of the Effect of siRNA-Mediated Interference with the Grid Gene Family

Following transfection with Cy3-siRNA for 24 h, red fluorescent signals were observed in the *spermatogonial cells* of *Cynoglossus semilaevis* (Figure 7B,C,E,F). After 48 h of siRNA transfection, qPCR detected the interference effect on *grid1* and *grid2* expression. The interference effect for *grid1* exceeded 70% (Figure 7A), while for *grid2*, it surpassed 60% (Figure 7D). These results indicate that the corresponding siRNA sites for *grid1* and *grid2* exhibit significant interference effects, providing a basis for subsequent downstream gene expression analysis.

### 3.5. Comparison of the Grid Gene Family RNAi Transcriptome Sequencing Data with the Cellular Genome

#### 3.5.1. Results of the *Grid* Gene Family Sequencing Data Quality Preprocessing

Following comparison with the reference genome of *Cynoglossus semilaevis* testicular cells, the average GC content of the *grid1* gene RNA interference (RNAi) samples was found to be 45.85%. The average Q20 quality score was 98.42%, the average Q30 quality score was 95.21%, and the error rate was 0.0125%. These sequencing quality metrics were sufficient for subsequent analyses (Table 2). Similarly, for the *grid2* gene RNAi samples, the average GC content was 46.82%, the average Q20 quality score was 98.38%, the average Q30 quality score was 95.08%, and the average error rate was 0.0125%. These metrics also indicated high sequencing quality suitable for further analysis (Table 3).

#### 3.5.2. Analysis of DEGs

In the experimental group treated with *grid1*-siRNA and its control group (*grid1*-NC), 99 differentially expressed genes (DEGs) were identified using the DEGseq software (https://bioconductor.org/packages/DEGseq/ accessed on 20 October 2024) with thresholds set at *p*-value < 0.05 and |log2 fold change| ≥ 1. Among these DEGs, 61 were up-regulated and 38 was down-regulated. Meanwhile, in the *grid2*-siRNA experimental group and its control group (*grid2*-NC), 623 DEGs were identified, with 495 up-regulated and 128 down-regulated (Figure 8). Clustering analysis results indicated a clear distinction between the experimental and control groups for both *grid1* and *grid2*, suggesting well-clustered samples and the feasibility of proceeding with further experiments (Figure 9).

#### 3.5.3. GO Function Annotation, GO Enrichment Analysis of Transcriptomic DEGs in Spermatogonial Cells After *Grid1*, *Grid2* Knockdown

GO function annotation and enrichment analyses were conducted for the grid1-siRNA and *grid1*-NC groups, as well as for the *grid2*-siRNA and *grid2*-NC groups, to explore the potential functions of DEGs after interfering with *grid1* and *grid2* in *Cynoglossus semilaevis* testicular cells. GO functional annotation after *grid1* knockdown showed that DEGs were primarily involved in biological processes such as catalytic activity, binding, protein-containing complexes, cellular anatomical entities, response to stimuli, metabolic processes, bioregulation, and cellular processes (Figure 10A). In contrast, GO annotation after *grid2* knockdown revealed that DEGs were mainly associated with cellular processes, metabolic processes, biological regulation, response to stimuli, developmental processes, multicellular organismal processes, and inter-organismal biological processes (Figure 10B).

GO functional enrichment analysis following *grid1* knockdown indicated that DEGs were significantly enriched in biological functions such as cell division, DNA polymerase complex, DNA replication, intracellular organelles, cytosolic organelles, and membrane-bound organelles (Figure 11A). After *grid2* knockdown, DEGs were enriched in DNA metabolism, DNA replication, cellular macromolecule metabolism, cell cycle, cell division, nucleic acid metabolism, cellular response to DNA damage, DNA repair, cellular stress response, and the metabolism of nucleotide-containing compounds (Figure 11B).

#### 3.5.4. KEGG Signaling Pathway Enrichment Analysis of DEGs in Cynoglossus Semilaevis Testicular Cells

The KEGG signaling pathway enrichment analysis of differentially expressed genes (DEGs) in the *grid1*-siRNA experimental group compared to the *grid1*-NC group indicated that numerous DEGs were notably enriched in 20 signaling pathways, including DNA replication, pyrimidine metabolism, purine metabolism, and cell cycle (Figure 12A). Similarly, the KEGG analysis for DEGs in the *grid2*-siRNA experimental group versus the *grid2*-NC group revealed significant enrichment in 20 signaling pathways, such as DNA replication, cell cycle, homologous recombination, mismatch repair, base excision repair, nucleotide excision repair, and cellular aging (Figure 12B).

#### 3.5.5. Effect of Expression of Growth-Related Genes After *Grid* Gene Family Disruption

To further investigate growth-related genes, mibp, shcbp1, sass6, cdca7, igf1, among others, were selected to construct a heat map illustrating the clustering of growth-related downstream genes (Figure 13). Subsequently, the genes shcbp1, cdca7, and sass6 were chosen for qPCR validation (Figure 14). Following the interference of the *grid1* gene, shcbp1 expression was up-regulated approximately 2-fold, cdca7 expression increased about 2-fold, and sass6 expression was up-regulated by roughly 2.2-fold. In contrast, when the *grid2* gene was disturbed, shcbp1 expression surged by approximately 3.9-fold, cdca7 expression by about 3-fold, and sass6 expression by around 3.8-fold. The qPCR results for these differentially expressed genes were consistent with the observed trends in the transcriptome data, thereby confirming the reliability of the transcriptome analysis of the *grid* gene family under perturbation. Additionally, the expression of igf1 was found to be down-regulated upon interference with both *grid1* and *grid2*, which captured our attention. Consequently, genes interacting with igf1 were selected for further qPCR validation, revealing an upregulation in gh expression (Figure 14).

## 4. Discussion

GWAS analysis has been extensively utilized in the study of *Cynoglossus semilaevis*, particularly in understanding the SSD mechanism. Key genes such as zbed1, nsd3, cdc45, klhl29, and smad4, predominantly located on the sex chromosomes, have been identified through GWAS analysis [20]. Additionally, candidate genes fblx19, plekha7, and nucb2 have been linked to disease resistance in *Cynoglossus semilaevis* through combined GWAS, Fst, and nucleotide diversity screening [21]. The csprf1l gene was identified via GWAS as being related to immune defense against *Vibrio harveyi* infection [22].

There is a notable female bias in the growth and sex of *Cynoglossus semilaevis*. A genome-wide association study (GWAS) was conducted on 500 individuals of each sex, selected from the same batch but of varying sizes, to analyze genes on the sex chromosomes. The study identified the *grid2* gene on the Z chromosome as correlating with body length, body weight, and BMI. Transcriptome analysis of this cohort revealed a negative correlation between *grid* gene expression and size in gonadal tissue. By integrating GWAS and transcriptome data, the *grid* gene family was identified as key to individual size differences in *Cynoglossus semilaevis*. Further analysis via qPCR confirmed that the expression levels of *grid* genes were inversely related to fish size.

The *grid1* gene has been associated with litter size [23] and fertility [24] in sheep, and with high fertility in Katahdin ewes [25]. *Grid1* gene expression is particularly prominent in neuroendocrine system organs such as the hypothalamus and brain [26]. In female rats, *grid1* may influence the onset of puberty by regulating gonadotropin-releasing hormone (GnRH) levels in the hypothalamus, RFamide-related peptide-3 (RFRP-3) levels, and progesterone (P4) [16]. In Yorkshire pigs, *grid1* is associated with synaptic function, cell proliferation, and energy production, with dlgap2 potentially affecting neuroendocrine regulation of reproductive processes. *Grid1* also appears to influence reproductive performance in horses [27]. Conversely, *grid2* plays a critical role in embryonic development during gestation in sows of the Sox breed [28], and is specifically associated with obsessive-compulsive disorder in females, but not males [29]. Copy number variants (CNVs) of *grid2* in Tibetan sheep are located in regions linked to growth and carcass quantitative trait loci (QTL) [30]. In small-tailed frigid sheep, *grid2* may promote luteinizing hormone (LH) secretion similar to the glutamate receptor PA1, thereby participating in ovulation [31]. Furthermore, the *grid2* gene has been linked to obesity [32] and body mass index [33] in humans, and it influences body weight in mice through neuromodulation of metabolism and energy homeostasis [34].

Disrupted differential genes such as shcbp1, whose ability to synergize with fgf13 in promoting cell proliferation through protein interactions is significantly enhanced, accelerate cell cycle progression [35]. Sass6, a core protein of the centriole biogenesis pathway [36,37], is a major structural component that constitutes the spoke and ensures that centrosomes assemble the nine-fold symmetric precursor [38], is essential for human cell and other organisms, and is essential for centromere formation in mouse sass6 which is required for centromere formation in the developing mouse embryo [39]. Cdca7, cell division cycle-associated7, plays a key role in maintaining DNA methylation patterns [40], deficiency can lead to genetically heterogeneous disorders characterized by neurodevelopmental delays [41], which may play a role in facilitating the organism’s growth and developmental process. qPCR and transcriptome results showed that the expression of the *grid* gene family was up-regulated after interfering with the expression of these genes, suggesting that there is a negative correlation between these genes and the individual size of the *Cynoglossus semilaevis*, which may affect the growth and development of the *Cynoglossus semilaevis*. In vertebrates and other organisms, individual size is mainly determined by differences in growth rate, which is regulated by the growth axis including growth hormone and the insulin-like growth factor secreted by the hypothalamus-pituitary-gonad and other tissues [42,43]. When the gh-igf1 axis is normal, the growth hormone-releasing hormone produced by the hypothalamus acts on the pituitary gland to produce the growth hormone gh, which accelerates the expression of the igf1 gene by stimulating the hepatic growth hormone receptor and thus, promotes the synthesis and release of igf1, which stimulates the growth of muscles, bones, and other organs and thus, promotes the growth of the body. Igf1 in turn feeds back to inhibit the release of the hypothalamic growth hormone, and the two are coordinated with each other and are indispensable. If the relationship between the two is disrupted, the growth and development of the organism will be affected [44,45,46]. Our results showed that the expression of igf1 and gh were opposite, and it was speculated that igf1 had a resistance to gh, and the regulation was deregulated by the knockdown of the *grid* gene family.

## 5. Conclusions

In this study, we investigated the effect of the *grid* gene family on the growth size of individual *Cynoglossus semilaevis* through a combination of GWAS sequencing, transcriptome sequencing, qPCR experiments, RNAi experiments, and transcriptome sequencing analysis after knocking down the *grid* gene family in spermatogonial cells. The results indicated that the *grid* gene family, identified through GWAS and transcriptome sequencing, showed a correlation with the body weight of *Cynoglossus semilaevis*. qPCR results demonstrated a negative correlation between the expression of the *grid* gene family and the fish size, with a slight difference in expression between males and females, where males exhibited slightly higher expression levels. The correlation of *grid2* gene expression with fish size was higher than that of the *grid1* gene. In a *Cynoglossus semilaevis* testicular cell line, RNAi-mediated transcriptome sequencing of the *grid* gene family revealed that the expression of the growth-related gene igf1 was down-regulated, while gh was up-regulated. This suggests that the *grid* gene family may influence the gh-igf1 axis to affect the growth of individual *Cynoglossus semilaevis*. We hypothesize that the *grid* gene family plays a significant role in the growth of *Cynoglossus semilaevis*, providing a foundation for further research into the growth size differences in this species.

## Figures and Tables

**Figure 1 animals-15-01130-f001:**
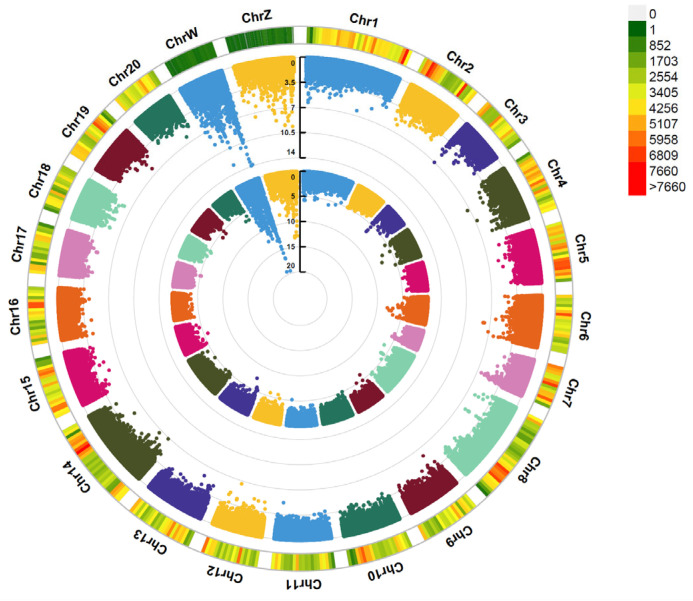
The Manhattan plot of the genome-wide association analysis (GWAS) for body weight and body length phenotypes in *Cynoglossus semilaevis*. Note: Figure 1 illustrates the distribution of SNPs across different chromosomes. The inner and outer circles represent the phenotypic associations for body weight and body length, respectively.

**Figure 2 animals-15-01130-f002:**
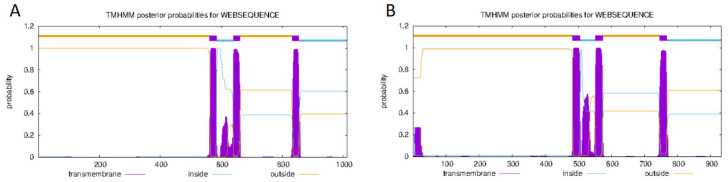
Analysis of transmembrane regions of the GluRD protein family of *Cynoglossus semilaevis.* Note: (**A**). Transmembrane region analysis of the *Cynoglossus semilaevis* GluD1 protein family; (**B**). Transmembrane region analysis of the *Cynoglossus semilaevis* GluD2protein family.

**Figure 3 animals-15-01130-f003:**
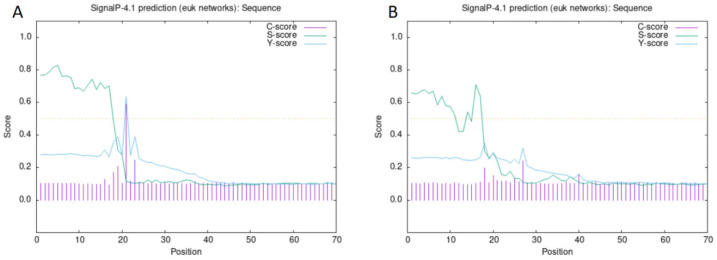
Signal peptide prediction and signal peptide shear site analysis of the GluD protein family of *Cynoglossus semilaevis*. Note: (**A**). Signal peptide prediction and signal peptide shear site analysis of the *Cynoglossus semilaevis* GluD1 protein family; (**B**). Signal peptide prediction and signal peptide shear site analysis of the *Cynoglossus semilaevis* GluD2 protein family.

**Figure 4 animals-15-01130-f004:**
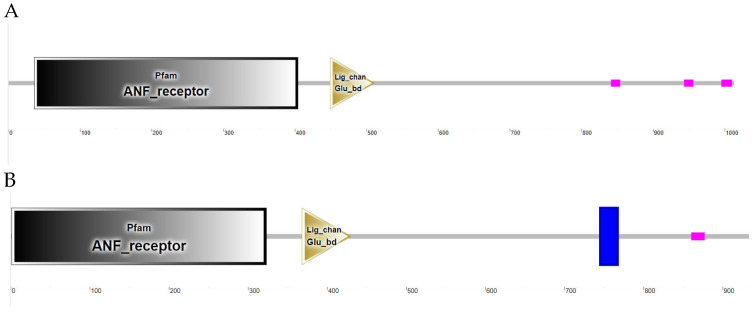
Structural domain analysis of the GluD protein family in *Cynoglossus semilaevis*. Note: (**A**). Structural domain analysis of *Cynoglossus semilaevis* GluD1 protein family; (**B**). Structural domain analysis of *Cynoglossus semilaevis* GluD2 protein family.

**Figure 5 animals-15-01130-f005:**
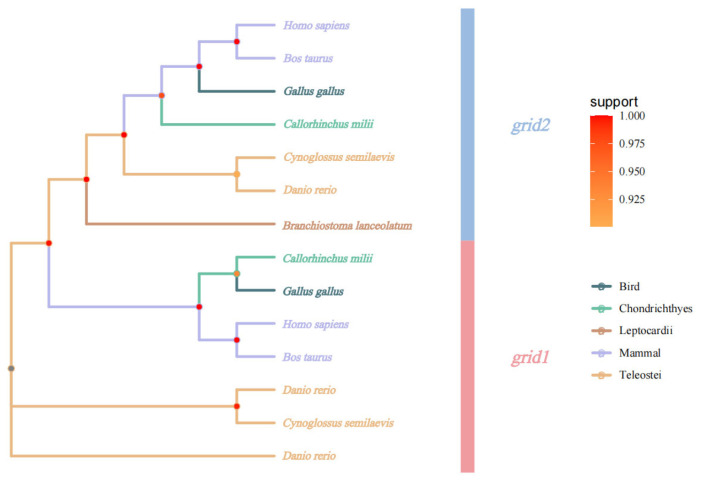
Phylogenetic evolutionary tree of *grid* gene family.

**Figure 6 animals-15-01130-f006:**
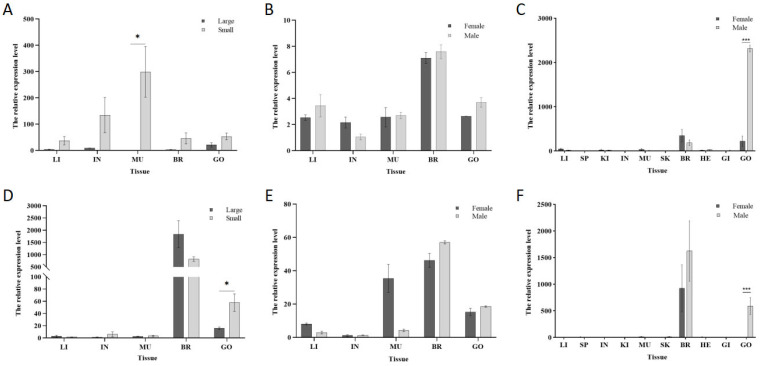
Analysis of the expression patterns of genes in the *grid* gene family in *Cynoglossus semilaevis*. Note: (**A**). Expression of *grid1* in each tissue in different sizes of the same sex; (**B**). Expression of *grid1* in each tissue in different sizes of the same sex; (**C**). Expression of *grid1* in whole tissues of different sizes of the male and female; (**D**). Expression of *grid2* in tissues in different sizes of the same sex; (**E**). Expression of *grid2* in tissues in different sizes of the same sex; (**F**). Expression of *grid2* in whole tissues of different sizes of the male and female. Expression in whole tissues of different sizes. * indicates significant difference (*p* < 0.05). *** indicates significant difference (*p* < 0.001).

**Figure 7 animals-15-01130-f007:**
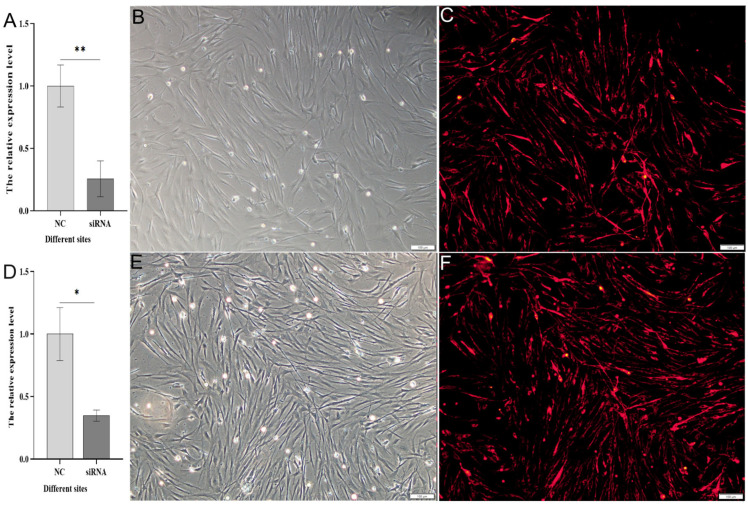
Expression changes of the *grid* gene family in *Cynoglossus semilaevis* after knockdown. Note: (**A**). Comparison of expression of NC after *grid1* knockdown with negative control; (**B**). Corresponding light microscopy images of (**C**); (**C**). Red fluorescence signal after transfection of *Cynoglossus semilaevis* testicular cells with Cy3-siRNA for 24 h when knocking down *grid1*. (**D**). Comparison of expression of NC after *grid2* knockdown with negative control; (**E**). Corresponding light microscopy images of (**F**); (**F**). Red fluorescence signal after transfection of *Cynoglossus semilaevis* testicular cells for 24 h when knocking down *grid2* with Cy3-siRNA transfection of *Cynoglossus semilaevis* testicular cells for 24 h after red fluorescence signal. * indicates significant difference (*p* < 0.05), ** indicates significant difference (*p* < 0.01).

**Figure 8 animals-15-01130-f008:**
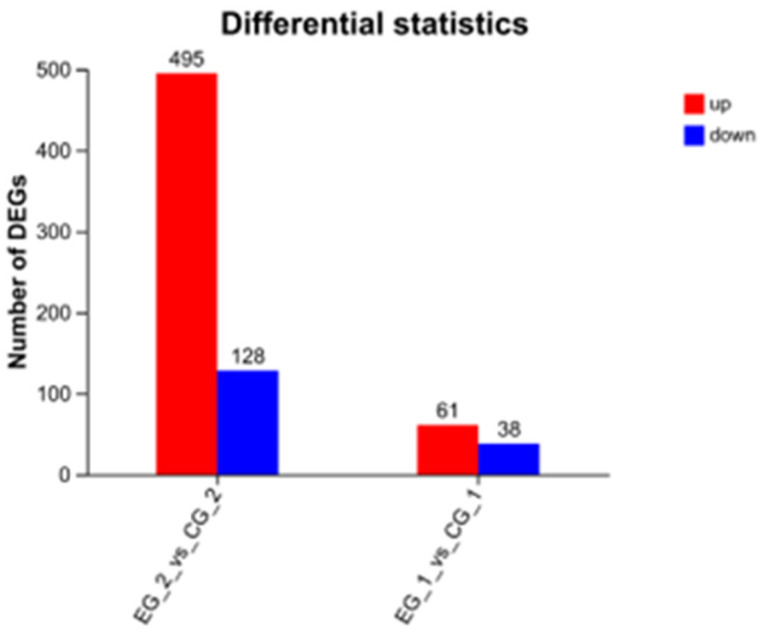
Changes in DEGs of the grid gene family. Note: DEGS change plots of the grid1-siRNA experimental group and the grid1-NC control group, the grid2-siRNA experimental group, and the grid2-NC control group, red represents up-regulated genes and blue represents down-regulated genes.

**Figure 9 animals-15-01130-f009:**
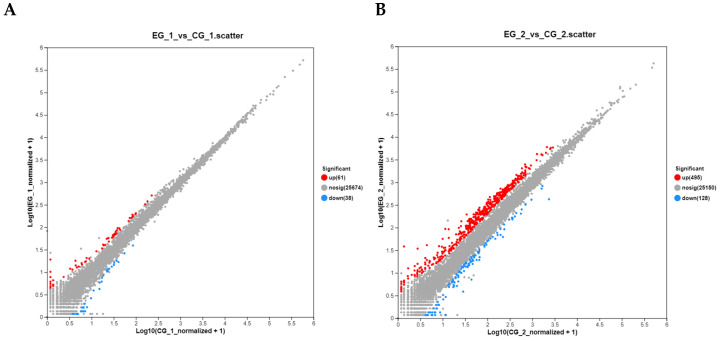
Volcano plot of DEGs. Note: (**A**). Volcano plot of *grid1*-siRNA experimental group and *grid1*-NC control group, red represents up-regulated genes and blue represents down-regulated genes; (**B**). Volcano plot of *grid2*-siRNA experimental group and *grid2*-NC control group, red represents up-regulated genes and blue represents down-regulated genes.

**Figure 10 animals-15-01130-f010:**
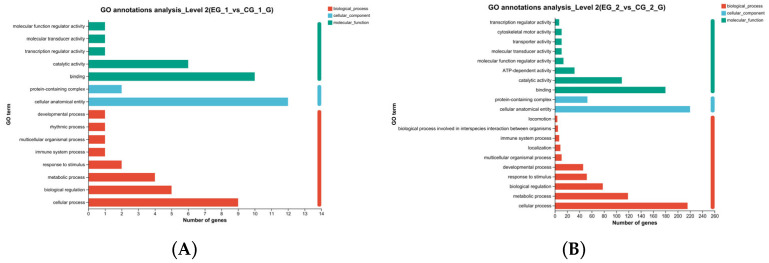
GO functional annotation of the *grid* gene family. Note: (**A**). *Grid1* GO function annotation; (**B**). *Grid2* GO function annotation. The left side is the description of the function of the gene, the horizontal axis represents count, the longer the bar, the more genes are enriched to the entry.

**Figure 11 animals-15-01130-f011:**
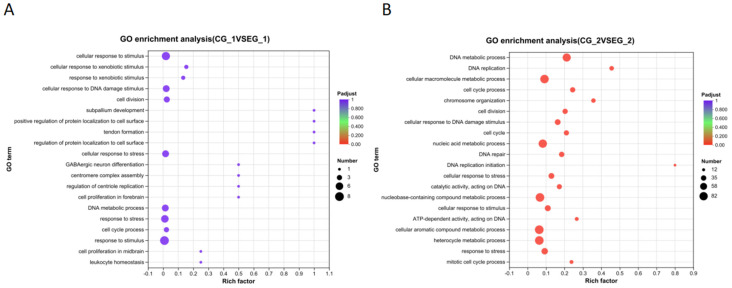
*Grid* gene family GO function enrichment bubble map. Note: (**A**). *Grid2* GO functional enrichment bubble chart; (**B**). *Grid1* GO functional enrichment bubble chart. The horizontal axis is the enrichment multiplicity, and the size of the dots represents the number of genes enriched for each function; the larger the dot, the greater the number of genes enriched above this GO entry, and the redder the dot, the more significant the corrected *p*-value.

**Figure 12 animals-15-01130-f012:**
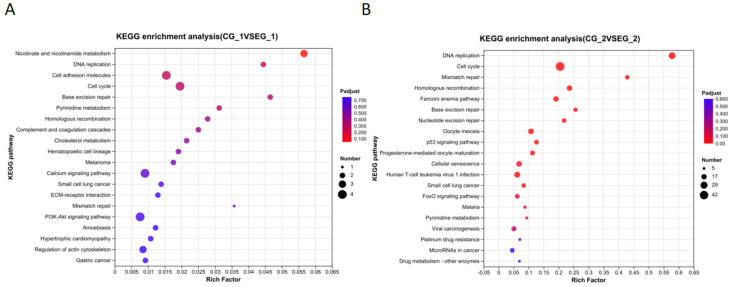
KEGG enrichment analysis bubble plot for the *grid* gene family. Note: (**A**). Bubble plot of KEGG enrichment analysis of *Grid1*; (**B**). Bubble plot of KEGG enrichment analysis of *Grid2*. The axis represents the name of the signaling pathway, and the horizontal axis represents Rich Factor. The size of the dots represents the number of genes enriched in each signaling pathway, and the larger the circle, the more genes are enriched in that signaling pathway. The color represents the *p*-value, the redder the dot, the more significant the corrected *p*-value.

**Figure 13 animals-15-01130-f013:**
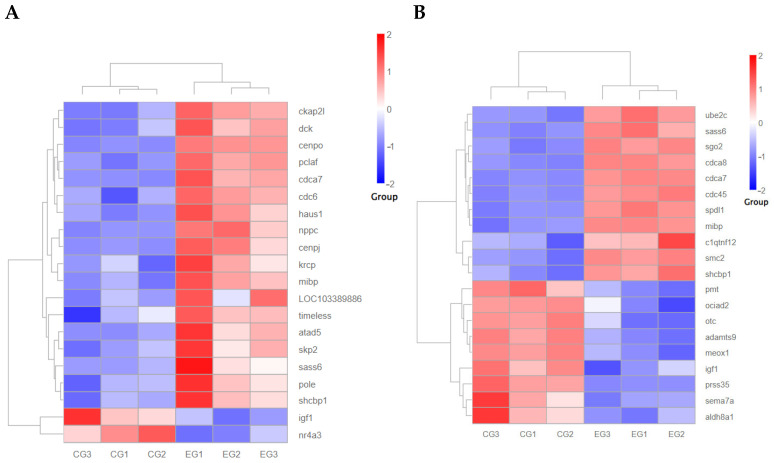
Clustering heatmap of downstream growth-related gene expression changes after disruption of the *grid* gene family expression. Note: (**A**). *Grid1* clustered heat map of downstream growth-related gene expression changes after gene expression disruption; (**B**). *Grid2* clustered heat map of downstream growth-related gene expression changes after gene expression disruption.

**Figure 14 animals-15-01130-f014:**
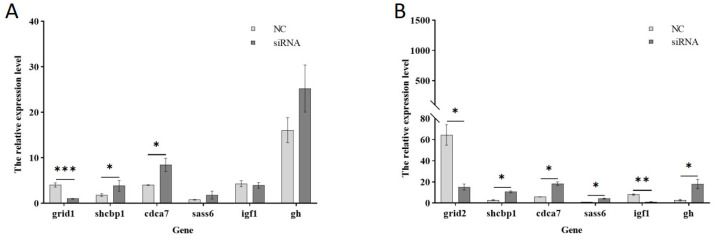
qPCR validation of downstream growth-related genes. Note: (**A**). *Grid1* gene expression changes; (**B**). *Grid2* gene expression changes. * indicates significant difference (*p* < 0.05), ** indicates significant difference (*p* < 0.01), *** indicates significant difference (*p* < 0.001).

**Table 1 animals-15-01130-t001:** The sequence of primers.

Primer	Sequence (5′~3′)	Purpose
*grid1*-F	GCGCTGAAATCCCGAGATGG	ORF verification
*grid1*-R	GTCTGTGGATGAATCTCCTGCT	ORF verification
*grid2*-F	TTTCTTCCATCCAGCCTGTGAG	ORF verification
*grid2*-R	AATCAGCCATATTCCCAGCAAC	ORF verification
Q-*grid1*-F	AACATTCTGGGCCAACCCAA	RT-qPCR
Q-*grid1*-R	AGCTCGCTTTCCGATCACTT	RT-qPCR
Q-*grid2*-F	TGATTTCCGCAACACCCACA	RT-qPCR
Q-*grid2*-R	AGGGCTACGGAAAAAGCCAC	RT-qPCR
*β-actin*-F	TTCCAGCCTTCCTTCCTT	RT-qPCR
*β-actin*-R	TACCTCCAGACAGCACAG	RT-qPCR
*gh*-F	ATCCACGCAGCCGGTTATAG	RT-qPCR
*gh*-R	CTCATGCTTGTTGTCGGGGA	RT-qPCR
*igf1*-F	ATGTCCATCTCTGCTCCGTC	RT-qPCR
*igf1*-R	GAAATAAAAGCCTCTCTCTCCAC	RT-qPCR
sex-F	CCTAAATGATGGATGTAGATTCTGTC	Gender detection
sex-R	GATCCAGAGAAAATAAACCCAGG	Gender detection
Q-*cdca7*-F	AGGAAATGCTCGCCAAACTG	RT-qPCR
Q-*cdca7*-R	TCTTCCTGCATGATCCCGGT	RT-qPCR
Q-*shcbp1*-F	GGCTACAAAGGAAATGCCAGG	RT-qPCR
Q-*shcbp1*-R	CAGGAGTCTGGTAGTGAGCAG	RT-qPCR
Q-*sass6*-F	GAACGAGTCGTCCATCAGAGA	RT-qPCR
Q-*sass6*-R	CTGACTCACCAAACGCTCCT	RT-qPCR

**Table 2 animals-15-01130-t002:** *Grid1* Sequencing data quality preprocessing results.

Sample	Raw Reads	Raw Bases	Clean Reads	Clean Bases	Error Rate (%)	Q20 (%)	Q30 (%)	GC Content (%)
Y1_1	48,615,382	7,340,922,682	48,102,788	7,177,069,845	0.0125	98.41	95.14	47.25
Y1_2	53,804,658	8,124,503,358	53,197,338	7,933,840,777	0.0125	98.42	95.2	46.72
Y1_3	47,486,224	7,170,419,824	46,941,306	6,986,664,819	0.0124	98.44	95.27	46.59
NC_1_1	51,427,878	7,765,609,578	50,797,030	7,578,762,047	0.0125	98.43	95.24	47.06
NC_1_2	51,475,210	7,772,756,710	50,868,992	7,577,515,371	0.0125	98.42	95.2	47.01
NC_1_3	45,453,332	6,863,453,132	44,943,864	6,709,376,354	0.0125	98.41	95.18	46.48

**Table 3 animals-15-01130-t003:** *Grid2* Sequencing data quality preprocessing results.

Sample	Raw Reads	Raw Bases	Clean Reads	Clean Bases	Error Rate (%)	Q20 (%)	Q30 (%)	GC Content (%)
Y2_1	47,141,102	7,118,306,402	46,616,214	6,960,704,393	0.0125	98.41	95.18	46.58
Y2_2	46,248,072	6,983,458,872	45,720,266	6,842,093,724	0.0126	98.34	94.94	46.89
Y2_3	48,204,732	7,278,914,532	47,684,778	7,130,821,946	0.0126	98.37	95.06	46.88
NC_2_1	45,554,842	6,878,781,142	45,025,372	6,721,019,003	0.0125	98.4	95.16	46.81
NC_2_2	45,094,196	6,809,223,596	44,599,126	6,662,407,768	0.0126	98.38	95.08	46.61
NC_2_3	49,930,426	7,539,494,326	49,407,854	7,390,924,078	0.0126	98.37	95.04	47.16

## Data Availability

All raw data have been deposited in the CNGB (China National GeneBank) Nucleotide Sequence Archive database (https://db.cngb.org/search/project) (accessed on 30 October 2024).

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
