# Peer review of "Molecular Mechanism of the Grid Gene Family Regulating Growth Size Heteromorphism in Cynoglossus semilaevis"

_animals, 2025, doi:10.3390/ani15081130_

Round 1
Reviewer 1 Report
Comments and Suggestions for Authors
The study ‘Molecular Mechanism of the Grid Gene Family Regulating Growth Size Heteromorphism in Cynoglossus semilaevis’ by Yaning Wang and colleagues is devoted to the investigation of the role of the grid family genes (grid1 and grid2) in growth differences in sea flounder (Cynoglossus semilaevis). GWAS, RNA sequencing (transcriptome analysis), qPCR, and RNA interference experiments on gonadal (spermatogonial) cell culture were used. Overall, the paper is logically structured and the results are very tightly supported by a combination of methods. The topic is relevant because the problem of differences in growth rates in Cynoglossus semilaevis is important both from the point of view of fundamental biology and, obviously, for aquaculture. I believe that the article can be published in the journal Animals after correcting a number of possible inaccuracies:
1. Section 2.4 states 350 individuals for GWAS, but the abstract and section 4 mentions 500 females for GWAS and transcriptome analysis. The sample composition needs to be clarified.
2. The term ‘spermatheca cells’ is used in the article. For fish, it is common to speak of either ‘spermatogonial cells’ or ‘testicular cells’. The term ‘spermatheca’ is usually used for some invertebrates (insects, molluscs) where there is a special sac/chamber to store sperm. It is advisable to check the correctness of the term. If a testes cell line is meant, it is better to call the whole thing ‘testicular cell line’ or ‘spermatogonia cell line’.
3. GWAS has identified 1536597 loci associated with growth phenotypes. That's quite a lot. Clarify if this is the number of snips (SNPs) rather than significant loci?
4. What is meant by ‘whole tissues’ (Fig. 6)? Is this a pool of all tissues or something else?
5. Please improve the readability of the small labels in Figures 4 and 5.
6. Apparently ethics committee approval is required to publish a paper. I did not find references to such a document.
Author Response
Comments 1. Section 2.4 states 350 individuals for GWAS, but the abstract and section 4 mentions 500 females for GWAS and transcriptome analysis. The sample composition needs to be clarified.
Response 1: Thank you for your valuable comments and we apologize for our carelessness. We had a total of 350 samples. The wrong number was corrected in the manuscript.
Comments 2. The term ‘spermatheca cells’ is used in the article. For fish, it is common to speak of either ‘spermatogonial cells’ or ‘testicular cells’. The term ‘spermatheca’ is usually used for some invertebrates (insects, molluscs) where there is a special sac/chamber to store sperm. It is advisable to check the correctness of the term. If a testes cell line is meant, it is better to call the whole thing ‘testicular cell line’ or ‘spermatogonia cell line’.
Response 2: Thank you very much for your valuable comments. After verification, the term “sperm cells” mentioned in the article is not standardized, the correct term should be testicular cell lines, we have changed all the errors in the article from “sperm cells” to "testicular We have changed all the errors in the article from “sperm cells” to “testicular cells”.
Comments 3. GWAS has identified 1536597 loci associated with growth phenotypes. That's quite a lot. Clarify if this is the number of snips (SNPs) rather than significant loci?
Response 3:
Thank you very much for your valuable comments. All 1,536,597 SNP markers were used for GWAS analysis after whole-genome resequencing and quality control of the sequencing results. These markers are not all specifically related to growth traits.
Comments 4. What is meant by ‘whole tissues’ (Fig. 6)? Is this a pool of all tissues or something else?
Response 4: Thank you for your valuable comments. The whole organization is meant to be the entire organization of the half-smooth tongue sole, but Figures 6c and 6f do not reflect all of the organization, and I have deleted this sentence.
Comments 5. Please improve the readability of the small labels in Figures 4 and 5.
Response 5: Thank you for your valuable comments. The small labels in Figures 4 and 5 are blurred, which affects readability, I have readjusted the images to improve their readability based on your comments, and we hope the revised version meets the requirements of the journal.
Comments 6. Apparently ethics committee approval is required to publish a paper. I did not find references to such a document.
Response 6: Thank you for your valuable comments and we apologize for our carelessness. Due to our carelessness, we forgot to add the documented information approved by the Ethics Committee in the paper. As per your suggestion, we have added the Ethics Committee approved in the paper. All the experiments in this study were processed under the inspection of Yellow Sea Fisheries Research Institute’s animal care and use committee (Approval number, YSFRI-2023006 on 2/3/2023). Fishes were anesthetization with MS-222 before sample collection.
Reviewer 2 Report
Comments and Suggestions for Authors
The study investigates the molecular mechanism underlying growth size heteromorphism in Cynoglossus semilaevis, primarily influenced by female-biased growth differences. The research topic is within the scope of Animals. Researchers conducted a genome-wide association study (GWAS) and transcriptome analysis on 500 female individuals of varying sizes to identify differential genes. The methods used for the study are appropriate. Quantitative PCR (qPCR) confirmed the expression patterns of grid gene family members in different tissues, revealing a negative correlation with fish size and slightly higher expression in males. RNA interference (RNAi) was used to knock down grid1 and grid2 in spermathecal cells, followed by transcriptome sequencing to examine downstream gene expression changes. Key findings include upregulation of shcbp1, sass6, cdca7, and gh, alongside downregulation of igf1, suggesting an antagonistic relationship between gh and igf1 when grid genes are suppressed. The results are sound and interesting. This study highlights the role of the grid gene family in regulating growth through the GH-IGF1 axis, providing valuable insights into the genetic basis of growth heteromorphism in C. semilaevis. The findings contribute to aquaculture by offering potential genetic targets for optimizing growth performance in this species. In general, the manuscript is well written. With some revisions and clarifications (see below comments in detail), the manuscript could be accepted for publication.
Comments in detail:
- Comments in detail:
- L35: Leave a space between "body weight" and "[3, 4]."
- L45-46: Provide sources for the claim: "Our database mining efforts revealed that the grid genes are associated with individual size in semilaevis."
- L78: Specify the age of Cynoglossus semilaevis used in the study.
- L87 & L91: Clarify the meaning of "August-aged" and "April-aged fish."
- L104: Since this is based on "our unpublished research," include details on how the 350 fish were sequenced.
- L109-110: Identify the reference "Brodie A, Azaria JR, Ofran Y. 2016."
- L205: Clarify whether the "1,536,597 loci" were used for GWAS or if they were associated with growth.
- L203-208: Describe which chromosomes had the largest effect on phenotypic variation.
Author Response
Comments 1:L35: Leave a space between "body weight" and "[3, 4]."
Response 1: Thank you for your comments, we have revised the manuscript according to your comments by adding a space between “weight” and “[3,4].” A space has been added between “weight” and “[3,4].”, and the method has been modified to read weight [3, 4]. The same change was made to all the manuscript.
Comments 2:L45-46: Provide sources for the claim: "Our database mining efforts revealed that the grid genes are associated with individual size in semilaevis."
Response 2:
Thank you for your comments. We conducted a GWAS analysis using the weight of 350 female individuals as the phenotypic trait. By identifying SNP loci related to weight, we discovered the grid gene, which suggests that it is related to growth, and therefore, to the growth of tongue sole.
Comments 3:L78: Specify the age of Cynoglossus semilaevis used in the study. Our database mining efforts revealed that the grid genes are associated with individual size in semilaevis
Response 3: Thank you for your comments, we have made a change based on your comments to add the age of Cynoglossus semilaevis as one age.
Comments 4:L87 & L91: Clarify the meaning of "August-aged" and "April-aged fish."
Response 4: Thank you for your valuable comments and we apologize for our carelessness "August-aged" and "April-aged fish" Was our mark which help us to distinguish the different fishes. We have deleted the words in the manuscript.
Comments 5: L104: Since this is based on "our unpublished research," include details on how the 350 fish were sequenced.
Response 5:
Since our key point in the article is not GWAS, the specific method of genome resequencing is not listed in the Materials and Methods section in last version.
We added The Materials and Methods as follows in the revision: A genome-wide association study (GWAS) was executed to identify single nucleotide polymorphisms (SNPs) linked to the regulation of growth phenotypes. The applicant has previously conducted a preliminary GWAS analysis on 350 half-smooth tongue sole. The study population had a total length ranging from 23.0 cm to 47.2 cm, with an average of 34.7 cm, and a weight range from 77g to 799g, with an average of 308.6g. There was a significant individual variation. Genomic DNA was extracted from the tail fin of 350 individual with different body size using the TIANamp Marine Animal DNA Kit (Tiangen Biotech, Beijing, China). Post quality assessment, qualified DNA underwent whole-genome resequencing via T7 for genotyping. Genotype data underwent whole-genome quality control post-resequencing. SNPs with a call rate below 90%, minor allele frequency below 5%, or significant deviation from Hardy-Weinberg Equilibrium (P < 10^-5) were excluded. Missing genotypes were imputed using Beagle 3.31. Post-QC and imputation, the remaining markers were consolidated into a genotype dataset.
Comments 6: L109-110: Identify the reference "Brodie A, Azaria JR, Ofran Y. 2016."
Response 6: Thank you for your comments, based on your comments I have removed the incorrectly inserted words“Brodie A, Azaria JR, Ofran Y. 2016.” from the manuscript.
Comments 7:L205: Clarify whether the "1,536,597 loci" were used for GWAS or if they were associated with growth.
Response 7:
Thank you very much for your valuable comments. All 1,536,597 SNP markers were used for GWAS analysis after whole-genome resequencing and quality control of the sequencing results. These markers are not all specifically related to growth traits.
Comments 8:L203-208: Describe which chromosomes had the largest effect on phenotypic variation.
Response 8:
Thank you very much for your valuable comments. The GWAS results show that the highest number of trait-associated SNP loci were detected on the sex chromosomes (Z and W), and the highest phenotypic variation rate for female growth traits was observed.